# Association between sarcopenia and falls in Chinese older adults: Findings from the China health and retirement longitudinal study

Chunhua Yang[1,2], Tengfei Ye[2], Yan Gao[3]*

1 Jinzhou Medical University Graduate Training Base, the 960th Hospital of People's Liberation Army Joint Logistics Support Force, Jinzhou, Liaoning, China, 2 Department of Geriatrics, Ruijin Hospital Lu Wan Branch, Shanghai Jiaotong University School of Medicine, Shanghai, China, 3 Department of General Medicine, The 960th Hospital of People's Liberation Army Joint Logistics Support Force, Jinan, Shandong, China

* gaoyantianyu74@163.com

## Abstract

Falling has become a significant factor in the mortality of elderly people. Little is known about whether sarcopenia can be a risk factor for falls in older adults. This study aims to assess the association between sarcopenia and falls among older Chinese according to the updated diagnostic guidelines of the Asian Working Group on Sarcopenia 2019 (AWGS 2019). We used data from the 2011 baseline and 2015 follow-up survey of the China Health and Retirement Longitudinal Study (CHARLS). This study examined the relationship between sarcopenia status and falls through cross-sectional analysis. Cox proportional hazards regression models were conducted to investigate the effect of sarcopenia status on subsequent falls, with the report of hazard ratio (HR). A total of 5,337 participants aged at least 60 years (51.3% men; mean age 67.6 ± 6.3) were enrolled in this analysis from the CHARLS 2011. The study revealed that the prevalence of falls was significantly higher in the possible sarcopenia and sarcopenia groups compared to the no sarcopenia group, with rates of 15.8%, 19.4%, and 24%, respectively. Logistic regression was utilized to investigate the association between sarcopenia and falls. Both possible sarcopenia (OR: 1.22, 95% CI: 1.03–1.45) and sarcopenia (OR: 1.64, 95% CI: 1.23–2.19) were positively associated with higher odds of falls (all $p < 0.05$). During the 4 years of follow-up, 1490 cases (29.9%) with incident falls were identified. In the longitudinal analysis, individuals with diagnosed sarcopenia (HR: 1.32, 95% CI: 1.11–1.57) were more likely to have new-onset incident falls than their no-sarcopenia peers. Sarcopenia in the elderly is an independent risk factor for falls, with health screening and intervention reducing fall risk and improving quality of life.

**Data availability statement:** The datasets used in this study can be accessed from the official website of CHARLS (Chinese Health and Retirement Longitudinal Study) at https://charls.pku.edu.cn/en/.

**Funding:** This research was supported by the 2023 PLA General Logistics Department Health Care Special Project [grant number 23BJZ45] from Yan Gao and the 2023 Shanghai Huangpu District Research Project [grant number HLM202202] from Chunhua Yang.

**Competing interests:** The authors have declared that no competing interests exist.

## Introduction

China is currently one of the countries with the largest and highest proportion of an aging population [1]. It is predicted that the population aged 65 and above in China will reach a quarter of the total population by 2050 [2]. With the aging population, the social and economic problems caused by falls among the elderly are becoming increasingly prominent [3]. Falls are defined as any unexpected change in position that causes an individual to rest on a lower surface, such as the floor or ground [4]. According to data from the World Health Organization (WHO), 30% of people aged 65 and above and 50% of people aged 80 and above worldwide report falling at least once a year [5]. Falls are the second leading cause of accidental injury-related deaths worldwide [6]. Especially in China, falls have become the main cause of fractures, repeated hospitalizations, disabilities, functional limitations, and even death among the elderly population. Therefore, based on the need to prevent and reduce falls in the elderly, people have begun to study modifiable and monitorable factors such as sarcopenia.

Rosenberg first proposed in 1989 that sarcopenia is a syndrome characterized by muscle loss, decreased muscle strength, and decreased physical performance, which gradually worsens with age [7], which may lead to a series of adverse outcomes such as functional decline, weakness, and death [8], as well as being a public social problem faced by an aging society [9]. In 2014, the Asian Working Group for Sarcopenia (AWGS) consensus defined sarcopenia as age-related loss of muscle mass, plus low muscle strength, and/or low physical performance and specified cutoffs for each component [10]. According to this standard, research has found that 5.5% to 25.7% of the elderly population in Asia suffers from sarcopenia. Therefore, sarcopenia may be the main cause of falls in elderly people. Several studies have shown that physical inactivity has a significant impact on the progression of sarcopenia [11]. In addition, patients with sarcopenia are more prone to falls and fractures than those without sarcopenia [12].

With the increasing knowledge of sarcopenia and the promotion of healthy aging, AWGS 2019 (Asian Myopenia Working Group 2019) introduced "possible sarcopenia", defined as low muscle strength or low physical performance, which will help raise awareness of sarcopenia prevention and intervention in various medical institutions [13]. Considering the clinical significance of possible sarcopenia, further exploration is needed to determine whether there is a relationship between possible sarcopenia and falls.

Currently, most research on the relationship between sarcopenia and falls comes from Western countries. However, research on the elderly population in China, specifically on the potential link between sarcopenia and falls as assessed by the AWGS 2019 criteria, remains scarce, highlighting a critical gap in health and public health research. This study used nationally representative data from the 2011 and 2015 China Health and Retirement Longitudinal Study (CHARLS) to conduct cross-sectional and longitudinal analyses to evaluate the relationship between sarcopenia, its subcomponents, and falls in the elderly population of Chinese communities.

## Methods

### Study population and design

The data is sourced from the Chinese Longitudinal Study on Health and Retirement (CHARLS). This project is a nationally representative survey conducted by the National Development Research Institute of Peking University on families and individuals aged 45 and above in China [14]. The national baseline survey for the CHARLS began in 2011 and covered 28 provinces. Aimed to collect population and health-related data on middle-aged and elderly people. Participants were selected using a multistage, stratified, cluster probability sampling strategy. The details of the study design have been reported previously [1]. Participants were followed up every two years by repeating similar examinations, with five waves of national data to date: 2011, 2013, 2015, 2018, and 2020. Data from the 2018 and 2020 follow-up surveys were incomplete and missing some health data, such as follow-up physical and blood tests [15]. Therefore, the 2011 dataset (involving 17,713 participants) was selected as the baseline and combined with the 2015 dataset (involving 21,095 participants) to construct the cohort study for longitudinal analysis.

This study was divided into two sections. (1) In the cross-sectional analysis, we used data from the baseline in 2011. A total of 17,713 participants were interviewed in CHARLS 2011, 12,376 individuals were excluded because of missing data on sarcopenia (n = 1,572), no fall index (n = 754), no information about gender (n = 20), aged < 60 years and missing age information (n = 10,030), leaving 5,337 participants for cross-sectional analysis. (2) In the longitudinal analysis, we further excluded 1,793 participants without follow-up data in CHARLS 2015. Our final analytic sample included 4,977 participants who had fall information in CHARLS 2011 and were followed up in 2015. The detailed selection process is shown in Fig 1.

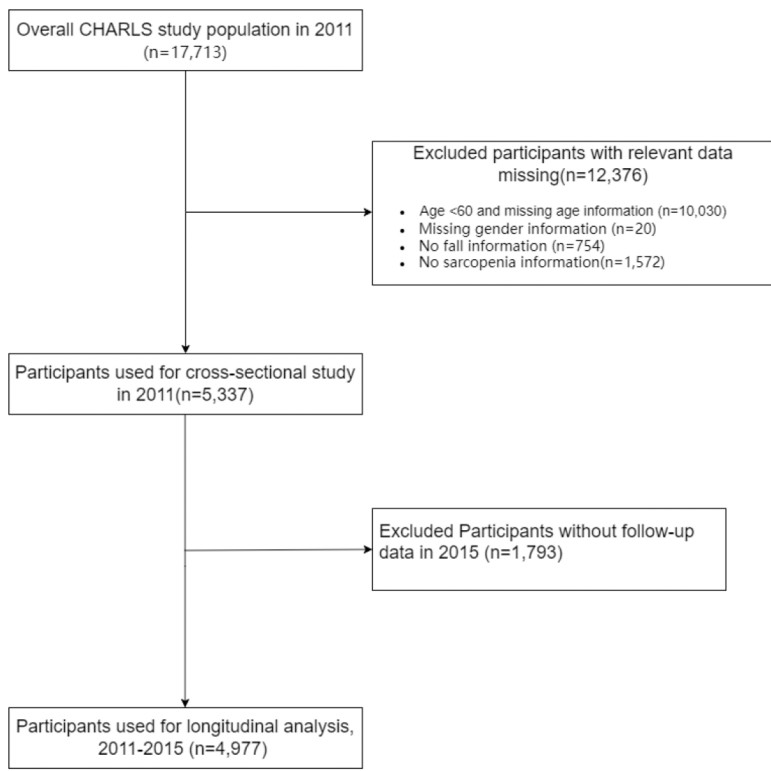

**Fig 1. Flowchart of the study cohort.**

CHARLS has been approved by the Ethics Committee of Peking University (No. IRB00001052–11015). All participants signed a written informed consent form at the time of participation, and their information was kept anonymous. All methods were performed according to the relevant guidelines and regulations. The investigation conforms to the principles outlined in the Declaration of Helsinki. The study datasets can be downloaded from the CHARLS homepage: http://charls.pku.edu.cn/en.

## Assessment of sarcopenia

Sarcopenia status was evaluated according to the AWGS 2019 algorithm, which incorporates three diagnostic components: muscle strength, appendicular skeletal muscle mass (ASM), and physical performance [13]. Muscle strength was assessed through handgrip strength measurement using a YuejianTM WL-1,000 dynamometer (Nantong Yuejian Physical Measurement Instrument Co., Ltd., Nantong, China) [1]. Participants performed maximal voluntary contractions with both dominant and non-dominant hands. Two measurements were taken for each hand with the dynamometer held at 90°flexion. The diagnostic thresholds for low grip strength were defined as < 28 kg for men and < 18 kg for women. Muscle mass was estimated by ASM using a validated anthropometric equation in Chinese residents [16,17], showing strong agreement with dual X-ray absorptiometry (DXA) [16,17]. ASM for the Chinese population was estimated using a physical measurement formula reported by a previous study [18]:

$$ASM = 0.193 \times weight(kg) + 0.107 \times height(cm) - 4.157 \times gender - 0.037 \times age(years) - 2.631$$

Where the weight was measured by the Omron™ HN-286 scale, and the height was measured by the SecaTM213 height meter. If male, gender was set to 1, otherwise to 0. The cutoff for defining low muscle mass was based on the sex-specific lowest 20% of the height-adjusted muscle mass ($ASM/Ht^2$) among the study population [17,19]. Low muscle mass was defined as $ASM/Ht^2$ values of <5.69 kg/m$^2$ in women and <6.88 kg/m$^2$ in men. Physical performance was assessed using the gait speed and the 5-time chair stand test, as described by Wu et al [19]. For gait speed, each participant was asked to walk a 2.5-m distance at a normal pace two times (there and back). The time to complete was recorded. The 5-time chair stand test measures the amount of time needed for the participants to rise continuously 5 times, keeping their arms folded across their chest from the height of the 47-cm chair. Low physical performance was defined according to the AWGS 2019 consensus as either a gait speed of < 1.0 m/s or a five-time chair stand test time of ≥ 12 seconds [13].

Sarcopenia is diagnosed when low muscle mass is combined with low muscle strength or low physical performance. Possible sarcopenia is identified by low muscle strength, with or without reduced physical performance. Individuals with all three components (low muscle mass, low muscle strength, and low physical performance) are classified as having severe sarcopenia [13]. Due to the small sample size of the severe sarcopenia group (n = 145, 2.7%), we combined it with the sarcopenia group for analysis to ensure statistical reliability, following established methodological precedents [20]. As a result, we combined those with severe sarcopenia into the sarcopenia group and divided all participants into three groups: no sarcopenia, possible sarcopenia, and sarcopenia.

## Falls

The primary outcome was fall accidents. Information on the incidence of falls and medical treatment resulting from falls was collected in the CHARLS [21]. Participants self-reported these events, which were captured in the 2011 baseline survey using the following question, "Have you had a fall in the past two years?" or captured in the 2013 and 2015 surveys using the following question, "Have you experienced a fall since your last visit?" Participants gave a binary response of either "yes" or "no".If the participant answered 'yes' to this question, they were then asked to specify the number of times falls led to medical treatment.

## Potential covariates

The collected data include age, gender, marital status (married or other), education level (primary school and below, middle school, and university and above), residential area (rural or urban), smoking and drinking status (yes or no), sleep time (hours/day), body height, weight, blood pressure, pulse, fall history, hip fracture history, and chronic disease (including hypertension, hyperlipidemia, diabetes, pulmonary disease, heart disease, stroke, kidney disease, digestive tract disease, and arthritis) history. Calculate the Body Mass Index (BMI) by dividing the weight in kilograms by the square of the height in meters. Hematological parameters include white blood cells, hemoglobin, total cholesterol, triglycerides, uric acid, C-reactive protein, creatinine, cystatin C, and glycated hemoglobin. Since the sarcopenia index (SI) has been assumed to be a potential indicator for the diagnosis of sarcopenia and may be associated with adverse outcomes [22], we calculated the SI. The specific formula employed is as follows: SI = serum creatinine/cystatin C × 100 [23].

## Statistical analysis

Firstly, descriptive statistics were employed to summarize the characteristics of participants at baseline. Normally distributed continuous variables were reported as mean ± standard deviation, while skewed continuous variables were presented as median with interquartile range (IQR). Categorical variables were expressed as frequencies and percentages. To assess differences in characteristics across the various sarcopenia status groups, the Chi-square test for categorical data and the Kruskal-Wallis test for continuous data were employed. A histogram was employed to illustrate the distribution of falls across the various sarcopenia status groups.

Secondly, a logistic regression analysis was performed to assess the relationship between possible sarcopenia, sarcopenia, its subcomponents, and falls in the cross-sectional study. Odds ratios (ORs) with 95% confidence intervals (CIs) were calculated for each model. In the longitudinal analysis, Cox proportional hazards models were used to determine the hazard ratios (HRs) with 95% CIs for the association between possible sarcopenia, sarcopenia, its subcomponents, and incident falls. Follow-up time was defined as the period from the last interview date to March 2015. Both cross-sectional and longitudinal analyses evaluated the association of low muscle mass, low handgrip strength, and low physical performance with falls. Variables with a univariate association ($p < 0.10$) or clinical relevance were included in multivariable logistic regression models. Three hierarchical models were constructed: Model 1 adjusted for age and sex; Model 2 added adjustments for marital status, education, BMI, smoking, and alcohol consumption; and Model 3 further adjusted for biochemical markers (Hgb, HbA1c, TG, UA, Cr, and CyC) and chronic diseases.

Thirdly, a sensitivity analysis was conducted on the cross-sectional and longitudinal analytical sample in order to evaluate the robustness of the relationship between sarcopenia status and falls and to evaluate the impact of different association inference models on our conclusions. The effect sizes and p-values calculated from all these models were documented and compared.

All statistical analyses were conducted using R Statistical Software (Version 4.2.2, http://www.R-project.org, The R Foundation) and Free Statistics Analysis Platform (Version 1.9.2, Beijing, China, http://www.clinicalscientists.cn/freestatistics). Statistical significance was defined as a two-sided p-value < 0.05.

## Results

### Baseline characteristics of participants in CHARLS 2011

A total of 5,337 participants aged 60 years and above were included in the 2011 baseline visit. Table 1 presents the general characteristics of sarcopenia participants based on different states. As shown in Table 1, the median age of the patients was 67.6 ± 6.3 years, with 2740 (51.3%) males and 2597 (48.7%) females. Among the participants, 1874 (35.1%) exhibited possible sarcopenia, while 1158 (21.7%) exhibited sarcopenia. Individuals with possible sarcopenia or sarcopenia were more likely to be older, female, unmarried, reside in rural areas, have a lower educational level, lower BMI, less

**Table 1. Baseline characteristics of the study population in 2011.**

| Characteristics | Total n = 5337 | No sarcopenia n = 2305 | Possible sarcopenia n = 1874 | Sarcopenia n = 1158 | p |
|---|---|---|---|---|---|
| Age, years | 67.6±6.3 | 65.5±4.8 | 67.7±6.1 | 71.6±7.1 | < 0.001 |
| Gender,n (%) | | | | | < 0.001 |
| Male | 2740 (51.3) | 1312 (56.9) | 899 (48) | 529 (45.7) | |
| Female | 2597 (48.7) | 993 (43.1) | 975 (52) | 629 (54.3) | |
| Marital status,n(%) | | | | | < 0.001 |
| Unmarried | 4269 (80.0) | 1965 (85.2) | 1499 (80) | 805 (69.5) | |
| Married | 1068 (20.0) | 340 (14.8) | 375 (20) | 353 (30.5) | |
| Resident arean(%) | | | | | < 0.001 |
| Rural | 2750 (79.9) | 1087 (73.4) | 994 (83) | 669 (87.8) | |
| Urban | 690 (20.1) | 393 (26.6) | 204 (17) | 93 (12.2) | |
| Educational level,n(%) | | | | | < 0.001 |
| Elementary school and below | 3021 (56.6) | 1031 (44.7) | 1143 (61) | 847 (73.1) | |
| Middle school and above | 2316 (43.4) | 1274 (55.3) | 731 (39) | 311 (26.9) | |
| Drinking, n(%) | 1312 (24.6) | 656 (28.5) | 424 (22.6) | 232 (20) | < 0.001 |
| Smoking, n (%) | 2324 (43.5) | 1041 (45.2) | 822 (43.9) | 461 (39.8) | 0.011 |
| Sleeptime, hours/day | 6.2±2.0 | 6.4±1.8 | 6.1±2.0 | 6.1±2.2 | < 0.001 |
| Height, cm | 156.2±17.5 | 158.0±12.0 | 155.6±13.5 | 153.6±28.6 | < 0.001 |
| Weight, kg | 56.5±11.6 | 61.0±10.3 | 55.7±11.1 | 48.7±10.4 | < 0.001 |
| BMI, kg/m$^2$ | 23.0±4.3 | 24.3±3.7 | 22.9±4.4 | 20.7±4.3 | < 0.001 |
| Incident fall, n (%) | 1005 (18.8) | 364 (15.8) | 363 (19.4) | 278 (24) | < 0.001 |
| Hip fracture, n(%) | 91 (1.7) | 32 (1.4) | 33 (1.8) | 26 (2.2) | 0.18 |
| Blood pressure | | | | | |
| Systolic,mmHg | 140.6±69.0 | 139.0±60.3 | 141.1±72.5 | 142.8±78.6 | 0.281 |
| Diastolic,mmHg | 75.0±11.9 | 76.1±11.2 | 74.9±12.2 | 73.2±12.5 | < 0.001 |
| Pulse beat, times/minute | 72.2±10.7 | 72.1±10.3 | 71.8±10.7 | 73.3±11.5 | < 0.001 |
| 5-time chair stand test, second | 11.8±5.0 | 9.1±1.8 | 13.0±5.2 | 15.5±5.7 | < 0.001 |
| Gait speed, m/s | 0.6 (0.5, 0.8) | 0.7 (0.5, 0.8) | 0.6 (0.5, 0.7) | 0.5 (0.4, 0.6) | < 0.001 |
| Handgrip strength,kg | 30.8±43.8 | 34.6±37.0 | 31.0±50.6 | 23.0±43.6 | < 0.001 |
| ASM | 16.3±4.5 | 17.7±4.0 | 15.9±4.1 | 14.1±5.1 | < 0.001 |
| ASM/Ht$^2$ | 6.7 (5.7, 7.4) | 7.1 (6.1, 7.7) | 6.6 (5.6, 7.2) | 6.0 (4.8, 6.7) | < 0.001 |
| Chronic disease | | | | | |
| Hypertension, n(%) | 1624 (30.5) | 729 (31.7) | 576 (30.8) | 319 (27.7) | 0.048 |
| Hyperlipidemia, n(%) | 528 (10.0) | 279 (12.3) | 175 (9.5) | 74 (6.5) | < 0.001 |
| Diabetes,n(%) | 362 (6.8) | 179 (7.8) | 118 (6.3) | 65 (5.6) | 0.033 |

*(Continued)*

**Table 1.** (Continued)

| Characteristics | Total n = 5337 | No sarcopenia n = 2305 | Possible sarcopenia n = 1874 | Sarcopenia n = 1158 | p |
|---|---|---|---|---|---|
| Pulmonary disease,n(%) | 730 (13.7) | 261 (11.4) | 267 (14.3) | 202 (17.5) | < 0.001 |
| Heart disease,n(%) | 804 (15.2) | 335 (14.6) | 291 (15.6) | 178 (15.4) | 0.641 |
| Stroke,n(%) | 149 (2.8) | 45 (2) | 51 (2.7) | 53 (4.6) | < 0.001 |
| Kidney disease,n(%) | 345 (6.5) | 146 (6.4) | 125 (6.7) | 74 (6.4) | 0.892 |
| Digestive tract disease,n(%) | 1210 (22.7) | 467 (20.3) | 461 (24.7) | 282 (24.4) | 0.001 |
| Arthritis, n(%) | 1978 (37.1) | 801 (34.8) | 729 (39) | 448 (38.8) | 0.009 |
| White blood cell,$10^9$/l | 6.3 ± 1.7 | 6.3 ± 1.6 | 6.3 ± 1.7 | 6.3 ± 1.8 | 0.902 |
| Hemoglobin, g/dl | 14.3 ± 1.9 | 14.5 ± 1.9 | 14.3 ± 2.0 | 13.9 ± 1.8 | < 0.001 |
| Total cholesterol, mg/dl | 194.2 ± 33.6 | 195.5 ± 33.1 | 194.1 ± 33.9 | 191.8 ± 34.0 | 0.01 |
| Triglycerides,mg/dl | 127.4 ± 83.5 | 132.1 ± 82.1 | 128.1 ± 90.9 | 117.1 ± 72.2 | < 0.001 |
| Uric Acid, mg/dl | 4.6 (3.9, 5.0) | 4.6 (4.1, 5.1) | 4.6 (3.8, 5.0) | 4.6 (3.8, 4.7) | < 0.001 |
| C-reactive protein, mg/dl | 1.8 (0.8, 3.2) | 1.8 (0.8, 3.2) | 1.7 (0.7, 3.2) | 2.0 (0.7, 3.2) | 0.085 |
| Creatinine, mg/dl | 0.8 (0.7, 0.9) | 0.8 (0.7, 0.9) | 0.8 (0.7, 0.9) | 0.8 (0.7, 0.8) | < 0.001 |
| Cystatin C,mg/l | 1.1 (1.0, 1.1) | 1.1 (1.0, 1.1) | 1.1 (1.0, 1.1) | 1.1 (1.1, 1.2) | < 0.001 |
| Glycated Hemoglobin,% | 5.3 (5.0, 5.4) | 5.3 (5.0, 5.4) | 5.3 (5.0, 5.4) | 5.3 (5.0, 5.3) | < 0.001 |
| SI | 75.7 (70.2, 77.3) | 75.7 (75.0, 80.3) | 75.7 (69.0, 76.6) | 75.7 (63.6, 75.7) | < 0.001 |

Abbreviations: BMI, body mass index; ASM, appendicular skeletal muscle mass; ASM/Ht², the skeletal muscle mass index (dividing ASM by the square of height); SI, sarcopenia index. Results are expressed as mean ± SD, or n (%). $p < 0.05$ was considered to be statistically significant.

sleep time per day, and a higher prevalence of chronic conditions (including hypertension, hyperlipidemia, diabetes, pulmonary diseases, stroke, digestive tract disease, and arthritis) compared to those without sarcopenia ($p < 0.05$). Furthermore, individuals with possible sarcopenia or sarcopenia exhibited lower hemoglobin, glycated hemoglobin, and SI levels ($p < 0.001$), as shown in Table 1.

## Cross-Sectional Association of Sarcopenia Status With Falls in CHARLS 2011

In the cross-sectional study conducted by CHARLS 2011, individuals with sarcopenia or possible sarcopenia exhibited a higher risk of falls than those without sarcopenia. The prevalence of incident falls in the total population, individuals without sarcopenia, those with possible sarcopenia, and those with sarcopenia were 1005 (18.8%), 364 (15.8%), 363 (19.4%), and 278 (24%), respectively (p for trend <0.001) (Table 1 and Fig 2). As shown in S1 Fig, a correlation exists between increased grip strength and a reduction in the risk of falls among older individuals. Nevertheless, as the duration of the

5-time chair-standing experiment increases, the risk of falls in older adults also decreases, as is shown in S2 Fig. Both variables exhibit a linear relationship.

The univariate logistic regression analysis of various variables and fall events revealed that participants with possible sarcopenia, sarcopenia, female, higher education level, rural living, less sleep time, chronic diseases, low hemoglobin, and low SI index exhibited a higher risk of falling events, as is shown in Table 2. In the multivariate logistic regression analysis, after adjusting for covariates, Table 3 demonstrated the association between sarcopenia states and falls. Individuals classified as having possible sarcopenia or sarcopenia demonstrated a significantly higher risk of falls, with adjusted odds ratios (ORs) of 1.22 (95% confidence interval (CI): 1.03–1.45; $p = 0.021$) and 1.61 (95% CI: 1.3–1.99; $p < 0.001$), respectively, compared to those without sarcopenia, after adjusting for all potential confounders.

**Longitudinal association between baseline sarcopenia status and falls at follow-up, 2011–2015**

During the follow-up period in CHARLS 2015, the proportion of incident falls in both the prevalence of possible sarcopenia and sarcopenic individuals increased to 612 (29%) and 472 (34.1%), respectively. This was still higher than that observed in the no sarcopenia individuals, as is shown in Fig 3. As shown in Table 4, the Cox proportional hazards model revealed that individuals with sarcopenia were more likely to experience an incident fall than those without sarcopenia (HR: 1.32; 95% CI: 1.11–1.57), after adjusting for potential covariates in models 1–3. However, no association was observed between possible sarcopenia and new-onset falls (HR: 1.11; 95% CI: 0.94–1.31).

In subgroup analysis in CHARLS 2011, we found that the interaction tests for arthritis were statistically significant($p = 0.001$), while the interaction tests for other variables (including age, gender, BMI, digestive system diseases, and diabetes) were not statistically significant($p > 0.05$), as is shown in Fig 4. During the follow-up in CHARLS 2015, we did not observe any interactions in the subgroups (all $p > 0.05$), as is shown in Fig 5.

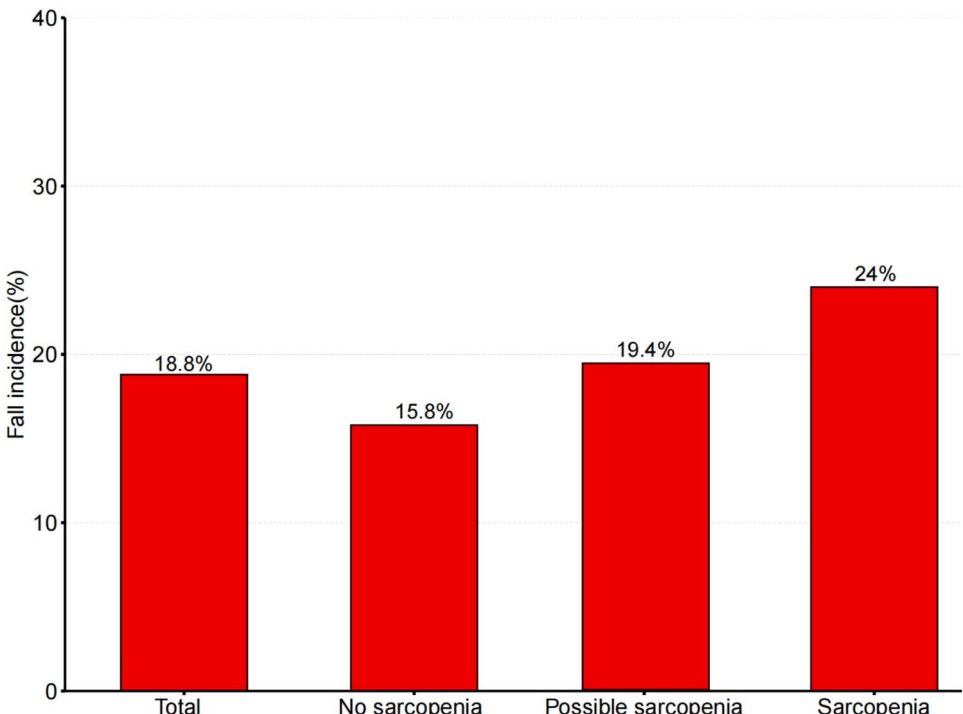

**Fig 2. Prevalence of falls in different groups in 2011.**

**Table 2. Logistic univariate analysis of the relationship between sarcopenia and falls in 2011.**

| Variable | OR_95 CI | *p*-value |
|---|---|---|
| Possible sarcopenia(vs.no sarcopenia) | 1.28 (1.09~1.5) | 0.002 |
| Sarcopenia(vs.no sarcopenia) | 1.68 (1.41~2.01) | <0.001 |
| Age | 1.02 (1.01~1.03) | 0.003 |
| Female(vs. male) | 1.5 (1.31~1.73) | <0.001 |
| Married(vs.unmarried) | 1.06 (0.9~1.26) | 0.49 |
| Education(vs.elementary school and below) | 0.68 (0.59~0.79) | <0.001 |
| Residence(urban vs.rural) | 0.67 (0.53~0.84) | 0.001 |
| Smoking (vs. never smoked) | 0.8 (0.69~0.92) | 0.002 |
| Drinking (vs. never drinking) | 0.99 (0.85~1.16) | 0.931 |
| Sleeptime | 0.88 (0.85~0.91) | <0.001 |
| Height | 0.99 (0.99~1) | <0.001 |
| Weight | 0.99 (0.98~1) | 0.001 |
| BMI | 1.03 (0.96~1.1) | 0.435 |
| Hypertension | 1.19 (1.03~1.38) | 0.021 |
| Hyperlipidemia | 1.36 (1.1~1.69) | 0.005 |
| Diabetes | 1.52 (1.19~1.94) | 0.001 |
| Pulmonary disease | 1.26 (1.04~1.52) | 0.019 |
| Heart disease | 1.27 (1.06~1.52) | 0.011 |
| Stroke | 1.49 (1.03~2.17) | 0.036 |
| Kidney disease | 1.47 (1.14~1.9) | 0.003 |
| Digestive tract disease | 1.57 (1.34~1.83) | <0.001 |
| Arthritis | 1.69 (1.47~1.95) | <0.001 |
| Hip fracture | 3.03 (1.98~4.63) | <0.001 |
| Hemoglobin | 0.95 (0.92~0.99) | 0.011 |
| Glycated Hemoglobin | 1.07 (0.97~1.17) | 0.159 |
| Triglycerides | 0.93 (0.86~1.01) | 0.068 |
| Uric Acid | 0.92 (0.86~0.98) | 0.007 |
| Creatinine | 0.45 (0.3~0.69) | <0.001 |
| Cystatin C | 0.74 (0.54~1.02) | 0.062 |
| SI | 0.99 (0.99~1) | 0.006 |
| 5-time chair stand test | 1.03 (1.02~1.04) | <0.001 |
| Gait speed | 0.6 (0.44~0.82) | 0.002 |
| Handgrip strength | 0.98 (0.97~0.98) | <0.001 |
| ASM/Ht$^2$ | 0.90(0.85~0.95) | <0.001 |

OR: Odds ratio; CI: confidence interval.

### Associations of subcomponents of sarcopenia with falls

In the cross-sectional and longitudinal analysis of the crude model, low muscle mass, low handgrip strength, and low physical performance were associated with falls. However, it was observed that only low grip strength and low physical performance were associated with falls in the fully adjusted model, as is shown in S1 and S2 Tables.

### Discussion

To the best of our knowledge, in the present study, we made a groundbreaking discovery by establishing a significant link between possible sarcopenia and the risk of falls in a large cohort of elderly patients. Our findings were based on data

**Table 3. Cross-sectional association between sarcopenia and falls in 2011.**

| Variable | n,% | Crude model | | Model1 | | Model2 | | Model3 | |
|---|---|---|---|---|---|---|---|---|---|
| | | OR 95%CI | *p*-value | OR 95%CI | *p*-value | OR 95%CI | *p*-value | OR 95%CI | *p*-value |
| No sarcopenia | 364 (15.8) | 1(Ref) | | 1(Ref) | | 1(Ref) | | 1(Ref) | |
| Possible sarcopenia | 363 (19.4) | 1.28 (1.09 ~1.5) | 0.002 | 1.22 (1.04 ~1.44) | 0.016 | 1.24 (1.05 ~1.46) | 0.013 | 1.22 (1.03 ~1.45) | 0.021 |
| Sarcopenia | 278 (24) | 1.68 (1.41 ~2.01) | <0.001 | 1.55 (1.28 ~1.88) | <0.001 | 1.64 (1.33 ~2.01) | <0.001 | 1.61 (1.3 ~1.99) | <0.001 |

OR: Odds ratio; CI: confidence interval. Crude model: no other covariates were adjusted. Model 1: We adjusted for age and sex. Model 2: We adjusted Model 1 + marriage status, education level, BMI, smoking, and drinking. Model 3: We adjusted Model 2 + Hgb, HbA1C, TG, UA, Cr, CyC, and chronic disease (including hypertension, hyperlipidemia, diabetes, pulmonary diseases, heart disease, stroke, kidney diseases, digestive tract disease, and arthritis).

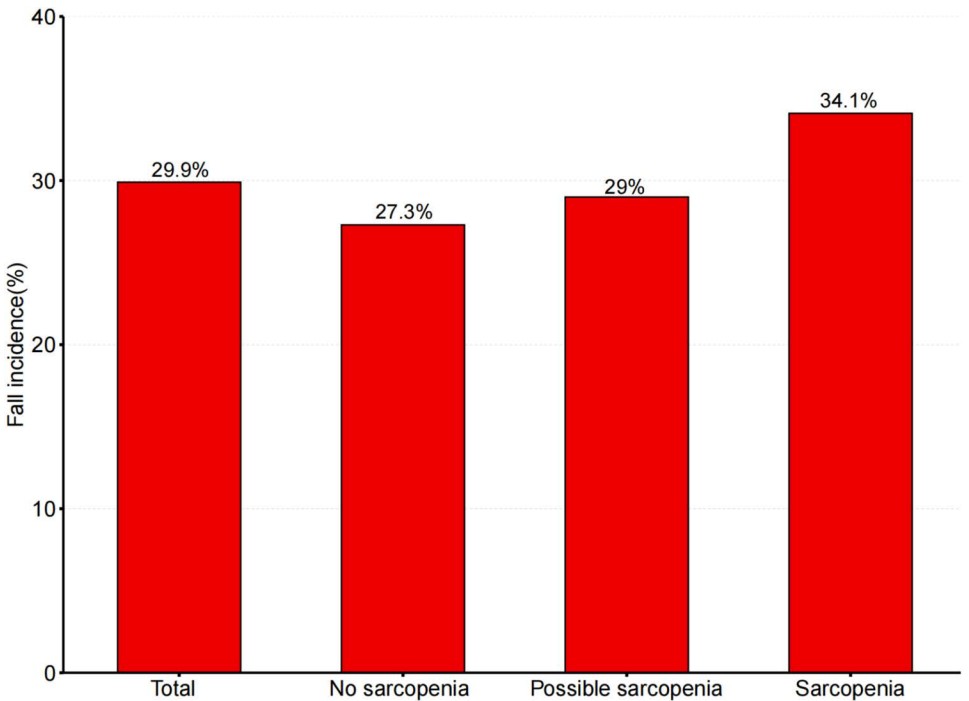

**Fig 3. Prevalence of falls in different groups, 2011 - 2015.**

from the China Health and Retirement Longitudinal Study (CHARLS). Furthermore, we investigated the impact of baseline sarcopenia on the probability of subsequent falls among older Chinese adults. The findings of the study revealed that both possible sarcopenia and sarcopenia were independently associated with an increased likelihood of falls in the cross-sectional analysis. Additionally, individuals with sarcopenia demonstrated a higher propensity for experiencing new-onset incident falls in our longitudinal analysis. Furthermore, we identified associations between low handgrip strength, diminished physical performance, and the occurrence of falls. These compelling results have significant clinical implications.

The identification of risk factors for falls and the development of effective intervention strategies represent public health priorities in numerous countries [24]. A number of physiological factors have been identified as potential risk factors for

**Table 4. Longitudinal analysis on sarcopenia and fall in the follow-up population, 2011 - 2015.**

| Variable | n,% | Crude model | | Model1 | | Model2 | | Model3 | |
|---|---|---|---|---|---|---|---|---|---|
| | | HR 95%CI | p-value | HR 95%CI | p-value | HR 95%CI | p-value | HR 95%CI | p-value |
| No sarcopenia | 406 (27.3) | 1(Ref) | | 1(Ref) | | 1(Ref) | | 1(Ref) | |
| Possible sarcopenia | 612 (29) | 1.07 (0.95 ~1.22) | 0.274 | 1.08 (0.95 ~1.22) | 0.256 | 1.08 (0.92 ~1.27) | 0.336 | 1.11 (0.94 ~1.31) | 0.209 |
| Sarcopenia | 472 (34.1) | 1.3 (1.14 ~1.49) | <0.001 | 1.26 (1.1 ~1.44) | 0.001 | 1.27 (1.07 ~1.5) | 0.006 | 1.32 (1.11 ~1.57) | 0.002 |

HR: Hazard Ratio; CI: confidence interval. Crude model: no other covariates were adjusted; Model 1: We adjusted age and sex. Model 2: We adjusted model 1 + marriage status, education level, BMI, smoking, and drinking. Model 3: We adjusted model 2 + Hgb, HbA1C, TG, UA, Cr, CyC, and chronic diseases (including hypertension, hyperlipidemia, diabetes, pulmonary diseases, heart disease, stroke, kidney diseases, digestive tract disease, and arthritis).

falls, including impaired gait, balance, and sensory function, as well as musculoskeletal and central nervous system damage. Psychological factors such as medication and its side effects, depression, and anxiety have also been linked to an increased risk of falls. Additionally, environmental factors such as living alone, slippery roads, obstacles, insufficient bathroom handrails, and improper walking aids [25] have been found to contribute to an elevated risk of falls. Collectively, these factors can increase the likelihood of a person falling. Our study found that the baseline incidence of falls in CHARLS 2011 was approximately 18.8%, which is consistent with previous research findings. A meta-analysis reported an average decline rate of about 18% [26], while Jun et al. found that after one year of follow-up, the incidence of falls in the elderly was approximately 16.8% [27]. The consistency of the fall incidence in this study with international research lends considerable credibility to our findings. Although the CHARLS database does not include specific indicators related to gait or gait disorders, such as the Tinetti Mobility Test, previous studies have highlighted the importance of gait and balance in fall risk assessment. For example, Cesari et al. demonstrated that the Tinetti Mobility Test is closely related to muscle mass and strength in non-institutionalized elderly individuals, suggesting that gait assessment could provide additional insights into fall risk [28]. While our study did not directly assess gait, the strong association between sarcopenia (a key determinant of muscle mass and strength) and falls further supports the importance of muscle-related factors in fall prevention.

Our findings are consistent with previous research indicating that age is positively correlated with the risk of falls. This suggests that with increasing age, the risk of falling also rises [29]. The process of aging is associated with a decline in physical activity, cognitive function, motor neuron function, and skeletal muscle mass, as well as an increase in sarcopenia. Furthermore, women are at a greater risk of falling than men. This phenomenon has also been observed in previous studies on elderly populations in the United States [30] and Canada [31]. The majority of falls occur within the home, with women being particularly susceptible to falls while performing kitchen tasks and household chores. Furthermore, post-menopausal elderly women experience a decrease in estrogen secretion [32], which results in imbalanced bone metabolism and an increased risk of osteoporosis. As age increases, there is a concomitant decrease in bone density, muscle strength, and physical function, all of which heighten the risk of falls. This subsequently impacts the quality of life and lifespan of elderly individuals.

In the cross-sectional study, the incidence rates of possible sarcopenia and sarcopenia in older people in the Chinese community were 35.1% and 21.7%, respectively, and the fall rates were 19.4% and 24%, respectively, both higher than in patients without sarcopenia. This conclusion is consistent with research in China [33] and abroad [34]. However, some foreign studies have not shown a significant association between falls and sarcopenia, which may be related to different measurement methods and diagnostic criteria for sarcopenia [34]. This study is based on the diagnostic criteria for possible sarcopenia and sarcopenia defined by the Asian Myopia Working Group in 2019. In addition, our research shows that

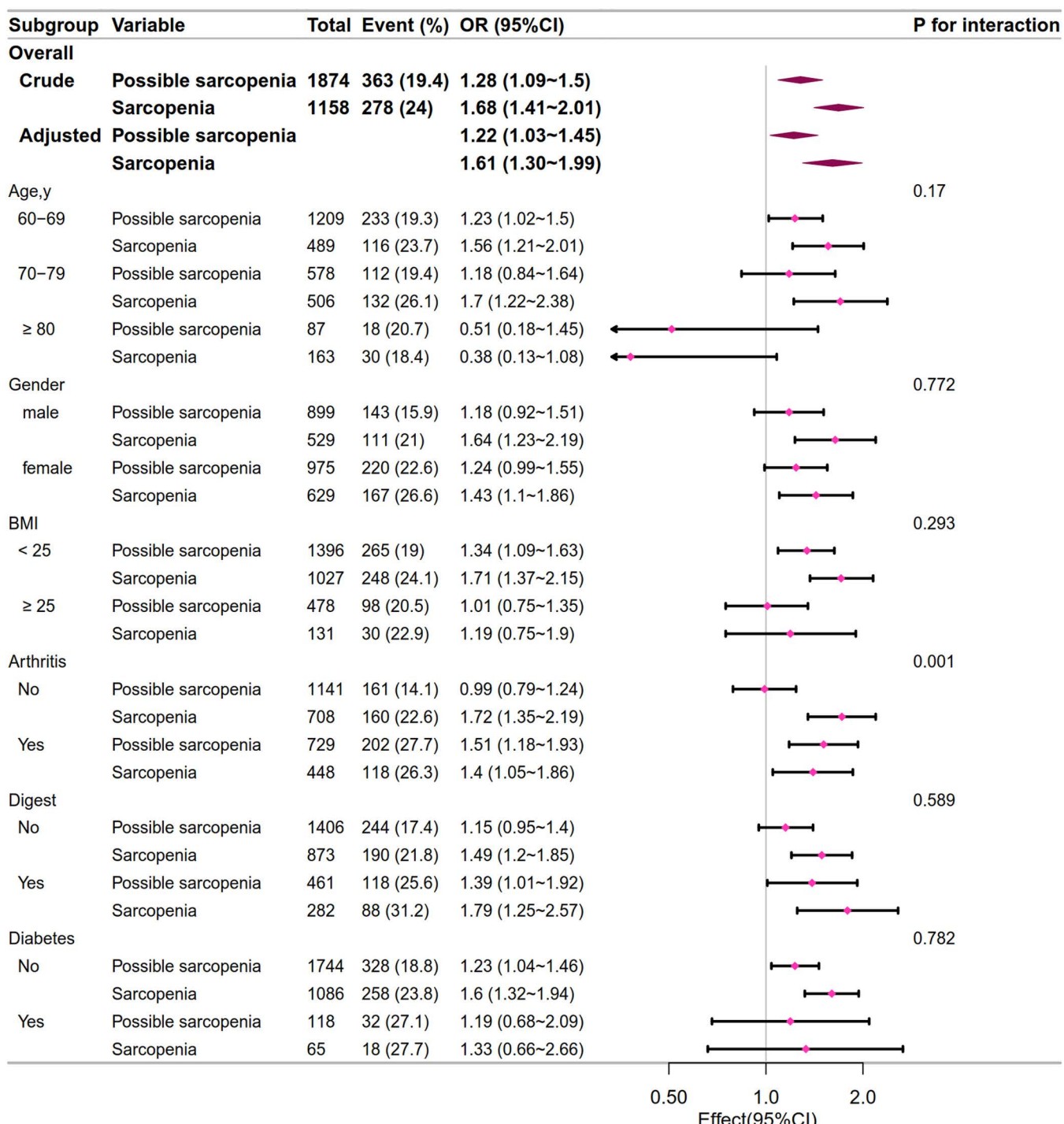

**Fig 4. The relationship between sarcopenia status and falls in the subgroup analysis in 2011.** The adjusted variables were age, gender, marriage status, education level, BMI, smoking, drinking, Hgb, HbA1C, TG, UA, Cr, CyC, and chronic disease (including hypertension, hyperlipidemia, diabetes, pulmonary diseases, heart disease, stroke, kidney diseases, digestive tract disease, and arthritis). OR: odds ratio.

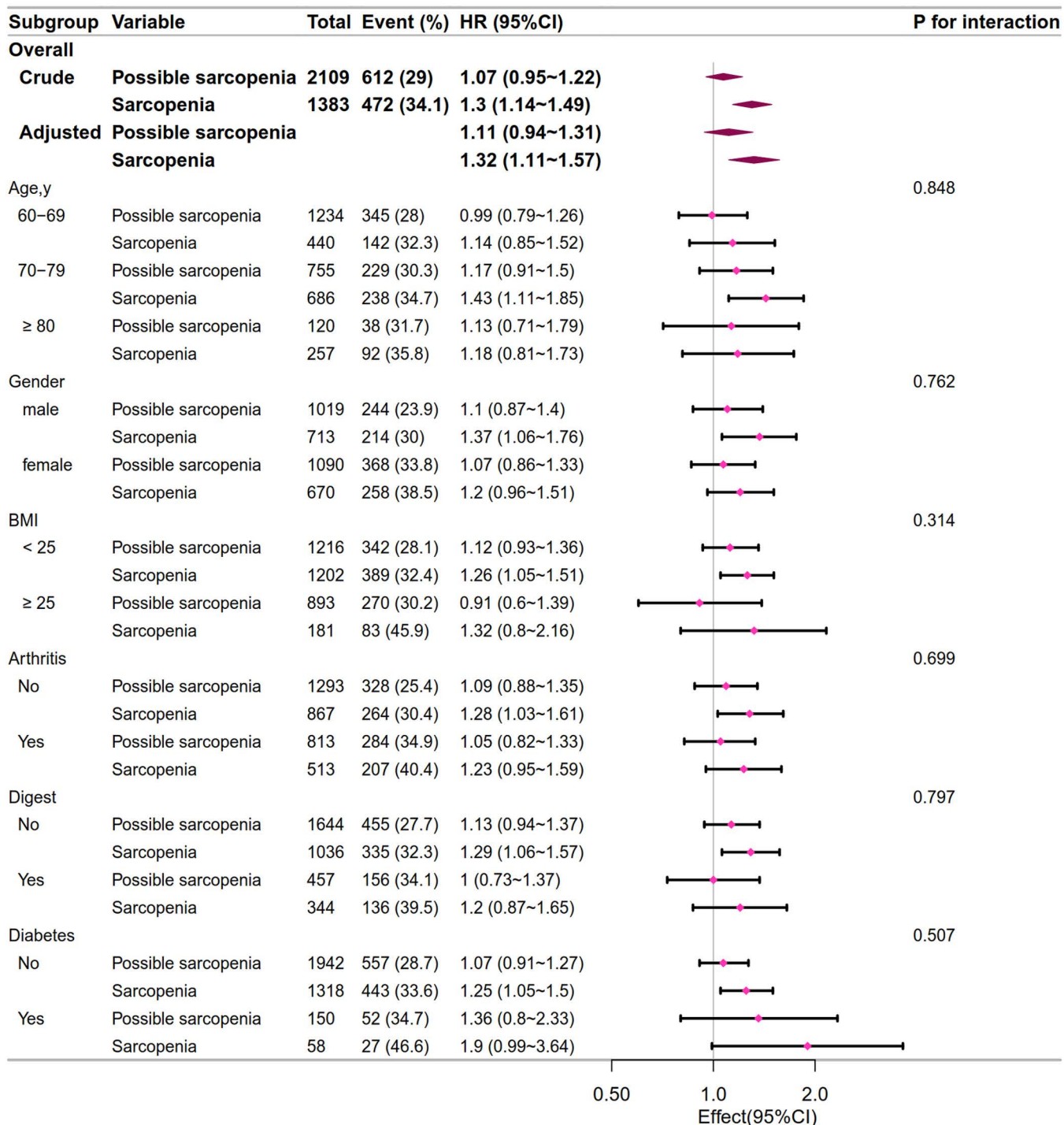

**Fig 5. The relationship between sarcopenia status and the subgroup analysis, 2011 - 2015.** The adjusted variables were age, gender, marriage status, education level, BMI, smoking, drinking, Hgb, HbA1C, TG, UA, Cr, CyC, and chronic disease (including hypertension, hyperlipidemia, diabetes, pulmonary diseases, heart disease, stroke, kidney diseases, digestive tract disease, and arthritis). HR: hazard ratio.

in horizontal multivariate logistic regression analysis, the risk of falling increases with the worsening of sarcopenia symptoms. However, after 4 years of observational follow-up, a higher incidence of falls was observed in sarcopenic patients. Previous studies have shown that the individual subcomponents of sarcopenia are associated with falls. Our study showed that low grip strength and low physical function were positively associated with increased fall risk after model adjustment. Conversely, low muscle mass was not correlated with falls. Physical performance is now considered to be a complex concept that includes muscle, central, and peripheral nervous system function, and balance [35]. Low physical fitness has become an important component in assessing vulnerability and severe muscle wasting, which is also associated with poor health [36]. Our findings are consistent with the consensus of the American Geriatrics Society and the British Geriatrics Society that muscle strength and physical function are important determinants of falls [37]. In addition, a Japanese study of 1,110 older people found that those who had fallen in the previous year had weaker grip strength and physical function than those who had not. However, the relationship between muscle mass and falls remains unclear [38]. This finding mirrors our research.

To contextualize our findings within sarcopenia research using the CHARLS dataset, two relevant studies are noteworthy. Wu et al. reported a sarcopenia prevalence of 21.7% among older Chinese adults, highlighting factors such as age, gender, and physical activity levels [39], which align with our findings. Liang et al. demonstrated that sarcopenia mediates the relationship between physical activity and falls [40], complementing our results by elucidating the interplay between these factors in fall risk among older adults with sarcopenia.

The mechanism behind the link between sarcopenia and falls can be explained in several ways. First, chronic inflammation may be the main cause. Our research shows that older people who fall are more likely to have various chronic diseases, and inflammatory cytokines can activate the ubiquitin-proteasome pathway, leading to muscle fiber degradation [41], resulting in reduced strength, function, and muscle mass, accelerating the development of sarcopenia, and increasing the risk of falls [42]. Second, in our study, we found that shorter sleep duration was associated with a higher risk of falls (OR: 0.88; 95% CI: 0.85–0.91) and a higher proportion of people with possible sarcopenia and sarcopenia. This is because insufficient sleep can lead to reduced secretion of insulin-like growth factor (IGF-1) and testosterone and increased levels of cortisol [43]. The reduction in IGF-1 and testosterone levels leads to reduced muscle protein synthesis, increased hydrolysis of skeletal muscle protein, and increased expression of muscle growth inhibitors. Meanwhile, an increase in cortisol levels can lead to muscle wasting [44]. Thirdly, aging is associated with a significant increase in insulin resistance, which in turn negatively affects muscle strength in older adults. Therefore, in the elderly population, early screening for possible sarcopenia and sarcopenic patients, followed by timely interventions such as enhanced physical activity, can not only directly improve insulin sensitivity but also indirectly modulate body composition. This dual effect contributes to the regulation of muscle and hepatic insulin sensitivity [45], thereby potentially delaying the progression of sarcopenia and reducing the risk of falls.

## Strengths and limitations

This study has several important strengths. First, this study uses data from the China Health and Retirement Longitudinal Study (CHARLS) to provide valuable insights into the association between sarcopenia and fall risk in older adults in China. Second, we examined the possible sarcopenia and fall risk and also analyzed the relationship between different subgroups of sarcopenia and falls, thus addressing a research gap in this area among the elderly population in China. Third, this study has the advantage of a large sample size, a nationally representative study population, and a longitudinal study design. This enhances the generalizability of our findings to the elderly population in China.

Despite the strengths of this study, several limitations should be acknowledged. First, variables such as falls and chronic diseases were self-reported, which may introduce recall bias. However, self-reported data on these outcomes are widely used and validated in aging studies. Rigorous quality control measures in CHARLS minimize reporting errors, ensuring the robustness of our findings despite potential biases. Second, while our study used anthropometric equations

to assess muscle mass instead of AWGS 2019's recommended DXA/BIA, this method has been validated for the Chinese population. Studies show strong agreement between ASM estimated by anthropometric equations and DXA measurements [46,47]. Although less precise than imaging, it provides a practical, reliable alternative for large-scale studies like CHARLS, balancing accuracy and feasibility. Third, the relationship between sarcopenia and falls may not be clear in the follow-up population, which may be related to the small sample size. Therefore, the representativeness of the research sample should be interpreted with caution, as a large number of participants were excluded. Fourth, although efforts were made to adjust for all relevant potential confounders in the multivariate model, the presence of unmeasured or unknown residual confounders (e.g., dietary factors, family income, and physical inactivity) cannot be completely excluded and may lead to an overestimation of the observed associations. Despite these limitations, our data effectively explores the association between sarcopenia and falls and adds further evidence to this area of research. Given the established association between sarcopenia progression and fall risk, healthcare providers should routinely evaluate physical function and muscle strength in older adults, monitor changes in sarcopenia status, and provide education on fall prevention and sarcopenia management. Such interventions may help reduce hospitalization costs for fracture-related care, as well as the risk of disability and mortality.

## Conclusion

In conclusion, this study suggests a positive association between sarcopenia and falls, as assessed by the AWGS 2019 criteria, in older people in China. This suggests that improving physical function, muscle strength, and muscle mass in people with sarcopenia may help reduce the incidence of falls. These findings emphasize the importance of regular physical function and muscle strength assessments in community-dwelling older adults to screen for sarcopenia and fall risks, supporting the implementation of evidence-based interventions, including nutritional optimization and physical activity enhancement. These efforts are essential for reducing fall-related mortality and disability in older adults, thereby holding significant clinical importance.

## Supporting information

**S1 Fig. Smooth curve fitting of incident falls in older patients in CHARLS 2011 with handgrip strength.**
(TIFF)

**S2 Fig. Smooth curve fitting of incident falls in older patients in CHARLS 2011 with the 5-time chair stand test.**
(TIFF)

**S1 Table. Cross-sectional association between components of sarcopenia and falls in 2011.**
(DOCX)

**S2 Table. Longitudinal association between components of sarcopenia and falls, 2011–2015.**
(DOCX)

## Acknowledgments

This study is based on the China Health and Retirement Longitudinal Study (CHARLS). We would like to thank the CHARLS research team, the field team, and every respondent for the time and effort that they have devoted to the CHARLS project. We thank all volunteers and staff involved in this research.

## Author contributions

**Conceptualization:** Chunhua Yang.

**Data curation:** Tengfei Ye.

**Formal analysis:** Chunhua Yang, Tengfei Ye.

**Funding acquisition:** Chunhua Yang, Yan Gao.

**Supervision:** Yan Gao.

**Writing – original draft:** Chunhua Yang.

**Writing – review & editing:** Yan Gao.

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
