## [Decision Letter · Decision Letter 0]

Dear Dr. yang,

Thank you for submitting your manuscript to PLOS ONE. After careful consideration, we feel that it has merit but does not fully meet PLOS ONE’s publication criteria as it currently stands. Therefore, we invite you to submit a revised version of the manuscript that addresses the points raised during the review process. 

We look forward to receiving your revised manuscript.

Kind regards,

Francesco Curcio, M.D., Ph.D.

Academic Editor

PLOS ONE

Journal Requirements:

https://www.frontiersin.org/journals/medicine/articles/10.3389/fmed.2021.755705/full

https://www.mdpi.com/2227-9067/8/12/1096

https://pubmed.ncbi.nlm.nih.gov/36058563/

In your revision ensure you cite all your sources (including your own works), and quote or rephrase any duplicated text outside the methods section. Further consideration is dependent on these concerns being addressed.

“This research was supported by the 2023 PLA General Logistics Department Health Care Special Project [grant number 23BJZ45] from Yan Gao and the 2023 Shanghai Huangpu District Research Project [grant number HLM202202] from Chunhua Yang.”

**Additional Editor Comments: **

The manuscript could be improved by addressing methodological ambiguities, enhancing comparisons with previous studies, and refining language and terminology for clarity.

Reviewers' comments:

Reviewer's Responses to Questions

**Comments to the Author**

1. Is the manuscript technically sound, and do the data support the conclusions?

Reviewer #1: Yes

Reviewer #2: Yes

2. Has the statistical analysis been performed appropriately and rigorously?

Reviewer #1: Yes

Reviewer #2: Yes

3. Have the authors made all data underlying the findings in their manuscript fully available?

Reviewer #1: Yes

Reviewer #2: Yes

4. Is the manuscript presented in an intelligible fashion and written in standard English?

Reviewer #1: Yes

Reviewer #2: Yes

Reviewer #1: Dear author, after carefully reading your manuscript, I congratulate you on your work. I have no comments to make. It is well developed, with a clear introduction and objectives, and a robust and replicable methodology.

Thank you for reviewing your manuscript.

Reviewer #2:

This study investigates the correlation between sarcopenia and falls among older adults in China using data from the 2011 baseline and 2015 follow-up surveys of the China Health and Retirement Longitudinal Study (CHARLS). The authors employ cross-sectional and Cox proportional hazards regression analyses to identify a statistically significant increase in fall prevalence among individuals with sarcopenia compared to those without. This research makes a valuable contribution by emphasizing the importance of early screening and intervention for sarcopenia to reduce incidences of falls.

The study addresses a critical public health issue with a robust dataset, providing meaningful insight into the relationship between sarcopenia and falls in an aging population. The manuscript is well-structured and delivers a thorough statistical analysis, crucial for understanding the dynamics between sarcopenia and fall risk. However, there are several issues that limit the study's value and raise concerns.

A significant limitation, as explicitly reported in the manuscript, is the use of anthropometric equations to estimate muscle mass. This method is not as precise as imaging techniques such as DXA or BIA, potentially affecting the accuracy of sarcopenia diagnoses. Additionally, the analysis relies on self-reported incidences of falls, which may introduce recall bias and affect the reliability of the results. Furthermore, it is uncertain whether these results are generalizable to non-Asian populations.

The criteria for selecting variables for the multivariate analysis are also not well described. It remains unclear whether an analysis of gait and gait disorders was conducted in these patients. In this regard, see and discuss “Tinetti mobility test is related to muscle mass and strength in non-institutionalized elderly people. Age (Dordr). 2016 Dec;38(5-6):525-533”

The discussion lacks comparisons with two other studies using the same CHARLS dataset but missing the longitudinal analysis: one by Wu et al. which examines the prevalence and associated factors of sarcopenia “ Sarcopenia prevalence and associated factors among older Chinese population: Findings from the China Health and Retirement Longitudinal Study. PLoS One. 2021 Mar 4;16(3):e0247617”, and another by Liang et al. which investigates the mediating role of sarcopenia in the association between physical activity and falls “The Mediating Role of Sarcopenia in the Association between Physical Activity and Falls among Chinese Older Adults: A Cross-Sectional Study. Healthcare (Basel). 2023 Dec 12;11(24):3146”. These comparisons could enrich the discussion by providing a broader context of sarcopenia research within the same population.

Minor issues in the manuscript include occasional awkward phrasing and grammatical errors that could hinder comprehension. There are also inconsistencies in terminology, such as alternating between "probable sarcopenia" and "possible sarcopenia" without clear distinctions, potentially confusing readers. Addressing these linguistic and terminological issues would enhance the clarity and professionalism of the manuscript.

**Do you want your identity to be public for this peer review?** For information about this choice, including consent withdrawal, please see our Privacy Policy

Reviewer #1: No

Reviewer #2: No

---

## [Author Response · Author response to Decision Letter 1]

19 Mar 2025

Dear Editors and Reviewers:

On behalf of my co-authors, we greatly appreciate the careful review and comments from both you and the reviewers. We believe that by implementing the suggested changes, we now have a stronger manuscript entitled “Association between sarcopenia and falls in Chinese older Adults: Findings from the China Health and Retirement Longitudinal Study” (ID: PONE-D-25-00109) for submission to PLOS ONE.

We have thoroughly reviewed the comments and implemented the necessary revisions, which we trust will meet with your approval. The revised sections are highlighted in red within the updated manuscript. These modifications do not affect the overall content or structure of the paper. Below, we have provided detailed point-by-point responses to each of the editor's and reviewers' comments, along with the corresponding revisions made to the manuscript. A comprehensive summary of the corrections and our responses to the feedback can be found in the Revision Report.

There are no conflicts of interest regarding this work. All authors have read the revised manuscript and approved its submission to PLOS ONE. Please do not hesitate to contact us if we can be of any further assistance. Thank you and best regards.

Yours Sincerely

Chunhua Yang

March 19, 2025

1 Jinzhou Medical University Graduate Training Base, the 960th Hospital of People’s Liberation Army Joint Logistics Support Force, Jinzhou, Liaoning, China

2 Department of Geriatrics, Shanghai Jiao Tong University Medical School Affiliated Ruijin Hospital Luwan Branch, Shanghai, China

Revision Report

First of all, I would like to express our sincere gratitude to the reviewers for their comments. These comments are all valuable and helpful for revising and improving our manuscript, as well as the important guiding significance to our researches. Based on suggestions, we have clarified the methodology, expanded the references, and strengthened the comparison with previous studies in the discussion. Additionally, we have refined ambiguous language and terminology in the manuscript to ensure clearer and more precise expression of our viewpoints. We have studied comments carefully and have made correction which we hope meet with approval. The revised sections in the manuscript are clearly highlighted in red text for easy reference. The summary of corrections and the responses to the editor's and reviewers' comments are listed below.

Responds to editors:

Comment 1:

Response 1:

Thank you for your comment. Following a detailed examination of the manuscript, we noticed that the file naming conventions for several figures did not adhere to the journal's style guidelines. We have now revised these to ensure full compliance with PLOS ONE's formatting standards.The revisions are extensive and not listed individually. Modified sections in the manuscript are highlighted in red for easy reference.

Comment 2:

We noticed you have some minor occurrence of overlapping text with the following previous publication(s), which needs to be addressed:

①https://www.frontiersin.org/journals/medicine/articles/10.3389/fmed.2021.755705/full

②https://www.mdpi.com/2227-9067/8/12/1096

③https://pubmed.ncbi.nlm.nih.gov/36058563/

In your revision ensure you cite all your sources (including your own works), and quote or rephrase any duplicated text outside the methods section. Further consideration is dependent on these concerns being addressed.

Response 2:

Thank you for your constructive suggestion. Upon a further thorough review of the manuscript, we identified minor textual overlap between the " Assessment of Sarcopenia" section in the Methods and references ① and ③. The authors have revised this section accordingly and added references ① and ③ to appropriately attribute the content. However, upon reviewing the content, methods, and discussion sections, we found no overlap between reference ② and our manuscript. We kindly request the editor to provide specific guidance on this matter so that we can make the necessary revisions. (①=[20]�③=[18] )

“Sarcopenia status was evaluated according to the AWGS 2019 algorithm, which incorporates three diagnostic components: muscle strength, appendicular skeletal muscle mass (ASM), and physical performance[13]. Muscle strength was assessed through handgrip strength measurement using a YuejianTM WL-1,000 dynamometer (Nantong Yuejian Physical Measurement Instrument Co., Ltd., Nantong, China)[1]. Participants performed maximal voluntary contractions with both dominant and non-dominant hands. Two measurements were taken for each hand with the dynamometer held at 90°flexion. The diagnostic thresholds for low grip strength were defined as < 28 kg for men and < 18 kg for women. Muscle mass was estimated by ASM using a validated anthropometric equation in Chinese residents[16,17], showing strong agreement with dual X-ray absorptiometry (DXA)[16,17]. ASM for the Chinese population was estimated using a physical measurement formula reported by a previous study[18]”

Page 6, lines 104-114;

“Due to the small sample size of the severe sarcopenia group (n=145, 2.7%), we combined it with the sarcopenia group for analysis to ensure statistical reliability, following established methodological precedents[20]. ”

Page 7, lines 130-133;

References:

“[18] Hu Y, Peng W, Ren R, Wang Y, Wang G. Sarcopenia and mild cognitive impairment among elderly adults: The first longitudinal evidence from CHARLS. J Cachexia Sarcopenia Muscle. 2022 Dec;13(6):2944–52. https://doi.org/ 10.1002/jcsm.13081 PMID: 36058563”

Page 25, lines 464-466;

“[20] Gao K, Ma WZ, Huck S, Li BL, Zhang L, Zhu J, et al. Association between sarcopenia and depressive symptoms in chinese older adults: evidence from the China health and retirement longitudinal study. Front Med. 2021;8:755705. https://doi.org/10.3389/fmed.2021.755705 PMID:34869454.”

Page 25-26, lines 471-474;

Comment 3:

Thank you for stating the following financial disclosure: 

“This research was supported by the 2023 PLA General Logistics Department Health Care Special Project [grant number 23BJZ45] from Yan Gao and the 2023 Shanghai Huangpu District Research Project [grant number HLM202202] from Chunhua Yang.”

Response 3:

The original funding was provided by two individuals; however, following discussions, it has now been revised to a single funder, namely Gao Yan, with the funding information as follows: " the 2023 PLA General Logistics Department Health Care Special Project [grant number 23BJZ45] from Yan Gao." We have specified the role of this funder as "methodology, supervision, funding acquisition, and writing—review & editing," and have added this information to the final section of the cover letter.

“This research was supported by the 2023 PLA General Logistics Department Health Care Special Project [grant number 23BJZ45] from Yan Gao

Role of Funder statement:

Yan Gao is the corresponding author who also undertakes the work of methodology, supervision, funding acquisition, and writing—review & editing in the research.”

the final section of the cover letter

Responds to the reviewer’s comments:

Reviewer #1

Comment 1:

Dear author, after carefully reading your manuscript, I congratulate you on your work. I have no comments to make. It is well developed, with a clear introduction and objectives, and a robust and replicable methodology. Thank you for reviewing your manuscript.

Response 1:

Thank you for your thorough evaluation and positive feedback on our manuscript. We are particularly grateful for the recognition of our work's clarity in introduction and objectives, as well as the robustness and replicability of our methodology. Our study represents the first large-scale longitudinal investigation of the sarcopenia-falls relationship in Chinese older adults using nationally representative data from CHARLS. The innovative application of the Asian Working Group for Sarcopenia criteria in this population provides a culturally relevant framework for assessing sarcopenia, while our robust statistical approach ensures strong reproducibility of findings across different aging populations. Most importantly, our findings provide practical implications for fall prevention strategies in elderly care, particularly through early identification and intervention of sarcopenia. We are encouraged by your supportive comments and believe this work will make a meaningful contribution to geriatric research in China. Once again, we deeply appreciate your time and valuable insights.

Reviewer #2

Comment 1:

This study investigates the correlation between sarcopenia and falls among older adults in China using data from the 2011 baseline and 2015 follow-up surveys of the China Health and Retirement Longitudinal Study (CHARLS). The authors employ cross-sectional and Cox proportional hazards regression analyses to identify a statistically significant increase in fall prevalence among individuals with sarcopenia compared to those without. This research makes a valuable contribution by emphasizing the importance of early screening and intervention for sarcopenia to reduce incidences of falls.

The study addresses a critical public health issue with a robust dataset, providing meaningful insight into the relationship between sarcopenia and falls in an aging population. The manuscript is well-structured and delivers a thorough statistical analysis, crucial for understanding the dynamics between sarcopenia and fall risk. However, there are several issues that limit the study's value and raise concerns.

Response 1:

We sincerely appreciate your insightful comments and positive evaluation of our work. We are particularly grateful for the recognition of our study's contribution to understanding the relationship between sarcopenia and falls in China's aging population. Acknowledging our methodological approach, including using CHARLS data and robust statistical analyses, is especially encouraging. We agree that early sarcopenia screening could significantly impact fall prevention strategies, and we thank you for highlighting this important implication of our findings. We sincerely appreciate the reviewer's valuable comments and will address each point to enhance the study's rigor and impact.

Comment 2:

A significant limitation, as explicitly reported in the manuscript, is the use of anthropometric equations to estimate muscle mass. This method is not as precise as imaging techniques such as DXA or BIA, potentially affecting the accuracy of sarcopenia diagnoses.

Response 2:

Although our study utilized anthropometric equations rather than the DXA or BIA techniques recommended by AWGS 2019, substantial evidence supports the validity of this approach. Previous studies have demonstrated strong concordance between ASM estimated by anthropometric prediction equations and DXA measurements through cross-validation. When DXA scanning is unavailable or impractical, anthropometric prediction equations serve as a reliable alternative for assessing appendicular skeletal muscle mass without compromising the diagnostic accuracy of sarcopenia. We have addressed this methodological consideration in the "Strengths and Limitations" section of our manuscript and have incorporated additional supporting references [46,47] to strengthen our methodological justification.

“Second, while our study used anthropometric equations to assess muscle mass instead of AWGS 2019 recommended DXA/BIA, this method has been validated for the Chinese population. Studies show strong agreement between ASM estimated by anthropometric equations and DXA measurements[46,47]. Although less precise than imaging, it provides a practical, reliable alternative for large-scale studies like CHARLS, balancing accuracy and feasibility. ”

Page 21, lines 373-377

References:

“[46] Villani AM, Crotty M, Cameron ID, Kurrle SE, Skuza PP, Cleland LG, et al. Appendicular skeletal muscle in hospitalised hip-fracture patients: development and cross-validation of anthropometric prediction equations against dual-energy X-ray absorptiometry. Age Ageing. 2014 Nov;43(6):857–62. https://doi.org/10.1093/ageing/afu106 PMID: 25049262

[47]Wen X, Wang M, Jiang CM, Zhang YM. Anthropometric equation for estimation of appendicular skeletal muscle mass in chinese adults. Asia Pac J Clin Nutr. 2011;20(4):551–6. https://doi.org/10.6133/apjcn.2011.20.4.13 PMID: 22094840 ”

Page 30, lines 556-562;

Comment 3:

Additionally, the analysis relies on self-reported incidences of falls, which may introduce recall bias and affect the reliability of the results.

Response 3:

We sincerely appreciate the reviewer’s insightful comment regarding the potential for recall bias in self-reported fall data. We acknowledge that self-reported measures may introduce some degree of recall bias; however, we would like to highlight several points supporting the validity and reliability of our findings based on the self-reported fall data in this study.

Firstly, self-reported fall data have been widely used in large-scale epidemiological studies, particularly in aging populations, and their validity has been well-documented. Studies have shown that self-reported falls in older adults exhibit a high concordance with objective measures, such as accelerometer data, especially when collected through structured interviews. For example, Ganz et al. demonstrated that self-reported falls in community-dwelling older adults had a sensitivity of 70%-80% compared to objective measures, supporting their utility in large cohort studies [1]. Similarly, Lusardi et al. found that self-reported fall events were reliable and valid for identifying fall risk in older populations, particularly when collected through standardized questionnaires [2].

Secondly, The China Health and Retirement Longitudinal Study (CHARLS) employs rigorous data collection protocols, including face-to-face interviews and standardized questions, to minimize recall bias. The use of a binary response (“yes” or “no”) to the question “Have you had a fall in the past two years?” or “Have you experienced a fall since your last visit?” is a well-established method for capturing fall incidents in longitudinal studies. Additionally, CHARLS incorporates follow-up questions to specify the number of falls requiring medical treatment, which further enhances the accuracy of the data. These methodological strengths have been validated in previous studies utilizing CHARLS data [3].

Thirdly, While recall bias cannot be entirely eliminated, the longitudinal design of CHARLS, with biennial follow-ups, reduces the risk of long-term recall errors. Furthermore, the large sample size and population-based nature of CHARLS help mitigate the impact of individual recall variability on the overall results. The consistency of our findings with prior literature on sarcopenia and fall risk also supports the robustness of our conclusions.

In summary, while recall bias cannot be entirely ruled out, the methodological rigor of CHARLS and the established validity of self-reported fall data support the reliability of our conclusions.

“First, variables such as falls and chronic diseases were self-reported, which may introduce recall bias. However, self-reported data on these outcomes are widely used and validated in aging studies. Rigorous quality control measures in CHARLS minimize reporting errors, ensuring the robustness of our findings despite potential biases.”

Page :21 lines:369-373

References:

[1] Ganz DA, Latham NK. Prevention of Falls in Community-Dwe

---

## [Decision Letter · Decision Letter 1]

Dear Dr. yang,

Thank you for submitting your manuscript to PLOS ONE. After careful consideration, we feel that it has merit but does not fully meet PLOS ONE’s publication criteria as it currently stands. Therefore, we invite you to submit a revised version of the manuscript that addresses the points raised during the review process.

We appreciate the comprehensive revisions and improvements made to the manuscript.

We encourage minor polishing of the English language (e.g., correcting typos such as "ealth" → "health", improving phrase flow) to further strengthen clarity.

We look forward to receiving your revised manuscript.

Kind regards,

Francesco Curcio, M.D., Ph.D.

Academic Editor

PLOS ONE

Journal Requirements:

Additional Editor Comments:

We appreciate the comprehensive revisions and improvements made to the manuscript. The methodology is robust, limitations are appropriately acknowledged, and previous reviewer concerns have been satisfactorily addressed.

We encourage minor polishing of the English language (e.g., correcting typos such as "ealth" → "health", improving phrase flow) to further strengthen clarity.

Reviewers' comments:

Reviewer's Responses to Questions

**Comments to the Author**

Reviewer #2: All comments have been addressed

2. Is the manuscript technically sound, and do the data support the conclusions?

Reviewer #2: Yes

3. Has the statistical analysis been performed appropriately and rigorously?

Reviewer #2: Yes

4. Have the authors made all data underlying the findings in their manuscript fully available?

Reviewer #2: Yes

5. Is the manuscript presented in an intelligible fashion and written in standard English?

Reviewer #2: Yes

Reviewer #2: The manuscript is really improved and all answers have been adresses. The manuscript merits to be published in PLOS ONE.

**Do you want your identity to be public for this peer review?** For information about this choice, including consent withdrawal, please see our Privacy Policy

Reviewer #2: No

---

## [Author Response · Author response to Decision Letter 2]

4 May 2025

ID: PONE-D-25-00109

Title of Manuscript: Association between sarcopenia and falls in Chinese older Adults: Findings from the China Health and Retirement Longitudinal Study

Dear editors

Thank you for your constructive feedback and recognition of our revised manuscript titled "Association between Sarcopenia and Falls in Chinese Older Adults: Findings from the China Health and Retirement Longitudinal Study". We sincerely appreciate the time and effort invested by the editors and reviewers in evaluating our work.

1.Language Polishing

In response to the editorial advice, the manuscript has undergone comprehensive English language editing to enhance clarity and academic rigor. Key revisions include typographical corrections (e.g., "ealth" → "health" in Line 34 of the Abstract), restructuring of ambiguous sentences (such as the paragraph in the Conclusion section), standardization of terminology (e.g., replacing "older people" with "older adults" ), and formatting adjustments to align with PLOS ONE guidelines (e.g., units and punctuation). All modifications are highlighted in the "Revised Manuscript with Track Changes" file for transparency.

2. Reference Updates

In accordance with PLOS ONE’s data policy, we reviewed the reference list and replaced two citations:

1)Original Reference 2: United Nations. (2019). World population prospects 2019: highlights. United Nations.Retreved May 1, 2024, from: https://www.un-ilibrary.org/content/books/9789210042352

Updated Reference 2: Zhou M, Wang H, Zeng X, Yin P, Zhu J, Chen W, et al. Mortality, morbidity, and risk factors in China and its provinces, 1990-2017: a systematic analysis for the Global Burden of Disease Study 2017. Lancet. 2019;394(10204):1145-1158. doi: 10.1016/S0140-6736(19)30427-1 PMID: 31248666

(Line: 413-416)

2)Original Reference 47: Wen X, Wang M, Jiang CM, Zhang YM. Anthropometric equation for estimation of appendicular skeletal muscle mass in chinese adults. Asia Pac J Clin Nutr. 2011;20(4):551–6. https://doi.org/10.6133/apjcn.2011.20.4.13 PMID: 22094840

Updated Reference 47: Kawakami R, Miyachi M, Tanisawa K, Ito T, Usui C, Midorikawa T, et al. Development and validation of a simple anthropometric equation to predict appendicular skeletal muscle mass. Clin Nutr. 2021;40(11):5523-5530. doi: 10.1016/j.clnu.2021.09.032I PMID: 34656948. (Line: 559-561)

No retracted articles are cited in the revised manuscript.

Submitted Files

Response to Editor: Detailed responses to editorial comments.

Revised Manuscript with Track Changes: All modifications highlighted.

Manuscript (Clean Version): Final unmarked version.

Data Availability

All data are publicly accessible via the CHARLS repository (http://charls.pku.edu.cn), consistent with the Data Availability Statement.

Should further revisions be required, please do not hesitate to contact us. We look forward to your favorable decision.

Yours Sincerely

Chunhua Yang1,2

May 4, 2025

---

## [Editor Report · Decision Letter 2]

Association between sarcopenia and falls in Chinese older Adults: Findings from the China Health and Retirement Longitudinal Study

PONE-D-25-00109R2

Dear Dr. chunhua yang ,

We’re pleased to inform you that your manuscript has been judged scientifically suitable for publication and will be formally accepted for publication once it meets all outstanding technical requirements.

Kind regards,

Francesco Curcio, M.D., Ph.D.

Academic Editor

PLOS ONE
---

## [Editor Report · Acceptance letter]

PONE-D-25-00109R2

PLOS ONE

Dear Dr. yang,

I'm pleased to inform you that your manuscript has been deemed suitable for publication in PLOS ONE. Congratulations! Your manuscript is now being handed over to our production team.

Kind regards,

on behalf of

Dr. Francesco Curcio

Academic Editor

PLOS ONE